# Laminarin, a Major Polysaccharide in Stramenopiles

**DOI:** 10.3390/md19100576

**Published:** 2021-10-15

**Authors:** Jichen Chen, Jianchao Yang, Hong Du, Muhmmad Aslam, Wanna Wang, Weizhou Chen, Tangcheng Li, Zhengyi Liu, Xiaojuan Liu

**Affiliations:** 1Guangdong Provincial Key Laboratory of Marine Biotechnology, STU-UNIVPM Joint Algal Research Center, Institute of Marine Sciences, Shantou University, Shantou 515063, China; 20jcchen1@stu.edu.cn (J.C.); hdu@stu.edu.cn (H.D.); aslam@stu.edu.cn (M.A.); 19wnwang@stu.edu.cn (W.W.); wzchen@stu.edu.cn (W.C.); tchli@stu.edu.cn (T.L.); 2Southern Marine Science and Engineering Guangdong Laboratory, Guangzhou 510000, China; 3Yantai Academy of Agricultural Sciences, Yantai 265500, China; yangjianchao_china@aliyun.com; 4Yantai Institute of Coastal Zone Research, Chinese Academy of Sciences, Yantai 264003, China; zyliu@yic.ac.cn

**Keywords:** glucan, laminarin, stramenopiles, microalgae

## Abstract

During the processes of primary and secondary endosymbiosis, different microalgae evolved to synthesis different storage polysaccharides. In stramenopiles, the main storage polysaccharides are β-1,3-glucan, or laminarin, in vacuoles. Currently, laminarin is gaining considerable attention due to its application in the food, cosmetic and pharmaceuticals industries, and also its importance in global biogeochemical cycles (especially in the ocean carbon cycle). In this review, the structures, composition, contents, and bioactivity of laminarin were summarized in different algae. It was shown that the general features of laminarin are species-dependence. Furthermore, the proposed biosynthesis and catabolism pathways of laminarin, functions of key genes, and diel regulation of laminarin were also depicted and comprehensively discussed for the first time. However, the complete pathways, functions of genes, and diel regulatory mechanisms of laminarin require more biomolecular studies. This review provides more useful information and identifies the knowledge gap regarding the future studies of laminarin and its applications.

## 1. Introduction

Eukaryotic photosynthetic microalgae are of global ecological importance. These microalgae acquire their photosynthesis ability with the establishment of plastids by endosymbiosis. The first endosymbiosis is speculated to have arisen 1–1.5 billion years ago [1]. During this event, a cyanobacterium was engulfed by a heterotrophic eukaryotic host, resulting in primary plastids surrounded by two membranes. There are three main lineages with primary plastids: Rhodophytes, Chlorophytes (including green plants) and Glaucophytes, forming together the Archaeplastida supergroup [2]. During the secondary endosymbiosis, the algae harboring the primary plastids were engulfed by another eukaryotic host and led to complex plastids surrounded by additional membranes [3]. There are four main lineages with Rhodophytes-derived complex plastids with four membranes: alveolates, stramenopiles, cryptophytes, and haptophytes, all of which belong together to chromalveolate group. Among these Rhodophytes-derived lineages, stramenopiles with complex plastids are also called heterokonts and contain diatoms, giant macroalgae such as kelps, and both photo-mixotrophic and heterotrophic species. In chromalveolate lineages, cryptophytes are the only member demonstrated to have retained their nucleomorph between the outer and inner chloroplast membrane pair; the nucleomorph was the retaining nucleus of the engulfed red algae [4].

During primary and secondary endosymbiosis, stramenopiles form a diverse lineage containing important photoautotrophic, mixotrophic, and heterotrophic taxa [5]. Diatoms are one of the most diverse groups in stramenopiles, containing at least 100,000 species, and contributing about 20% of annual global carbon fixation. Therefore, diatoms play a very important role in global biogeochemical cycles [6]. In diatoms, the fixed carbon is mainly used to synthesize polysaccharide, storing it in the vacuole. The vacuolar polysaccharide is a central energy metabolite and accounts for up to 50% of organic carbon in sinking diatom-containing particles; it will be released extracellularly and degraded by bacterium directly in the ocean [7]. Therefore, it is important for the carbon export from the surface ocean and carbon cycle [8]. It was found that these vacuolar polysaccharides are β-1,3-glucans, which are different from the storage polysaccharides (α-1,4-gulcans (starch)) in plastids of land plants and green algae [9]. β-1,3-glucans were mainly composed of glucose and also called laminarin, chrysolaminarin, or mycolaminarin depending on the algal species. β-1,3-glucan in brown algae was termed laminarin, in diatom *Phaeodactylum tricornutum*, Chrysophyte *Poterioochromonas malhamensis* and Eustigmatophyceae class *Nannochloropsis gaditana* was named chrysolaminarin and, in oomycetes, mycolaminarin [9,10,11,12,13]. These β-1,3-glucans are the main carbohydrate molecules in the ocean carbon cycle and carbon pool [8]. In addition, it was also reported that β-1,3-glucans possess various biological activities and functions in food, cosmetic and pharmaceuticals [10].

Owing to the important bioactivity of β-1,3-glucans, the development of microalgae-based microbial cell factories has gained wide attention in recent years. Usually, stress conditions are beneficial for the accumulation of high-value compounds, such as triacylglycerides (TAG) and chrysolaminarin [14]. Furthermore, genome editing tools are another breakthrough for the development of microalgae as cell factories. With the rapid development of biotechnology, shortly after the first sequenced *Thalassiosira pseudonana*, until now, nine diatoms’ genomes were deciphered [6]. The molecular tools for genetic engineering of diatoms were also developed, including gene silencing via RNAi, gene knockout via transcription activator-like effector nucleases (TALEN) and Crispr/Cas9 [15], plastid transformation and the episomes delivered by bacterial conjugation, and biolistic transformation and electroporation [14]. Additionally, genetic tools (e.g., electroporation transformation and Crispr/Cas9 gene editing) are also built up in non-diatom stramenopiles, such as in eustigmatophyte *Nannochloropsis oceanica* with the whole genome sequenced [5]. The sequenced genomes and the development of genetic tools are important for the study of genes’ functions and their regulatory mechanisms during the biosynthesis of chrysolaminari. They also play fundamental roles in producing large amounts of bioactive chrysolaminarin. Therefore, this review reports on recent findings regarding the species-specific features of laminarin, the proposed biosynthesis and catabolism pathway of laminarin, functions of some key genes and diel regulation of laminarin and summarizes the current knowledge of laminarin in stramenopiles.

## 2. General Features of Laminarin in Stramenopiles

### 2.1. Laminarin Structure and Composition

The structure was significantly various in diatoms, even between closely related organisms (Table 1). It was shown that the molecular weight of β-1,3-glucans ranged from 2 kDa to 40 kDa in stramenopiles. The degree of polymerization and degree of branching are different in stramenopiles, even in diatoms. This might explain the discrepancy of molecular weight in stramenopiles. In addition to the backbone β-1,3-glucose, the side chain is mainly composed of β-1,6-glucose (except with additional β-1,2-glucose in *Skeletonema costatum*, *Stauroneis amphixys,* and *Achnanthes longipes*, with additional β-1,3-glucose in *Stephanodiscus meyerii*, *Aulacoseira baicalensis* and *Chaetoceros muelleri*, and with 1-linked D-mannitol in *A. baicalensis* and *Nannochloropsis gaditana*). The usual structure of laminarin with the backbone β-1,3-glucose and the side chain β-1,6-glucose was shown in Figure 1. It was reported that the structural characterization of chrysolaminarin was affected by the culture conditions, such as the chrysolaminarin extracted from *Chaetoceros muelleri* and *Thalassiosira weissflogii* [16]. It was shown that the composition of chrysolaminarin in *Odontella aurita* mainly is glucose (82.23%) [17]. However, the composition of laminarin is also dependent on its purity. Usually, the extract of laminarin was contaminated by other carbohydrate. Moreover, the chrysolaminarin was also identified in haptophyte *Pleurochrysis haptonemofera*, the degree of polymerization and degree of branching are 203 and 1.5, respectively [18]. The great different structures of chrysolaminarin between stramenopiles and haptophytes might be related with the evolutional relationships of these two microalgae.

Regarding the structure of laminarin in brown algae, the backbone usually consists of β-1,3-glucoses with a small number (≤10%) of β-1,6-branching glucose residues [19]. The degree of polymerization is usually 15–40 with a molecular weight of 2–10 kDa [20]. According to the terminal ends, two types of laminarin are described: (1) M type, also named M-chains, where the backbone chain of the laminarin contain a D-mannitol residue at their reducing terminal; and (2) G type, also named G-chains, where the D-mannitol residue is replaced by glucose residue [21]. The percentage of these two types and the structure of laminarin were strongly relied on the species and environmental factors [22]. Moreover, the types of laminarin are dependent on the harvesting time of brown alga. Laminarin with 1.5 ratio of β-1,3: β-1,6-glucose and a molecular weight close to 5 kDa was found from Eisenia bicyclis collected in May [23]. However, in addition to the usual laminarin with 5 kDa, the high molecular weight laminarin (19–27 kDa) was also detected from the algae collected in July [19]. 

Several analysis methods of laminarin have already been established. For example, chrysolaminarin can be easily observed in the vacuole by aniline blue dye [32,33] or anti-β-1,3-glucan antibodies [34]. The monosaccharide composition of chrysolaminarin can be measured by complete acid hydrolysis and the GC-MS analysis methods [17]. The structure of chrysolaminarin can be analyzed by different methods, such as 1H nuclear magnetic resonance (NMR), 13C NMR, Fourier-transform infrared (FTIR) spectra, glycosyl linkage analysis, and size exclusion chromatography [35].

### 2.2. Laminarin Content

The content of β-1,3-glucans can be extracted or measured by different procedures, such as hot water extraction [10], alkali extraction [36], phenol sulfuric acid assay [37], 3-methyl-2-benzothiazolinone hydrazone (MBTH) reducing sugar assay [25], enzymatic hydrolysis [7,8], and high-performance size-exclusion chromatography (HPSEC) method [38]. Among these methods, hot water extraction could achieve the highest amount of β-glucans with high purity and low cost [10]. However, the optimal extraction method is related with the structure and sources of laminarins [39].

The content of laminarin was relied on algal species, growth phases, the extracted methods, etc. In brown algae, laminarin content varied from 20 to 50% of dry weight [21], while the content of chrysolaminarin ranged from 0.4 to 55% (dry weight) in other microalgae (Table 1). It was reported that chrysolaminarin can reach up to 20–30% of dry weight during the exponential growth phase of diatoms and increase up to 80% during the stationary phase [11]. The contents of chrysolaminarin in diatoms *Chaetoceros affinis* and *S. costatum* were low in the exponential phase and rapidly increased in the stationary phase [40]. Additionally, the content was also stimulated by the high light intensity [17,40]. The extraction methods are also important for the content of chrysolaminarin. The value of chrysolaminarin extracted via the MBTH reducing sugar assay was lower than that from dilute sulfuric acid method, which contained other non-glucose containing carbohydrates [25,41].

Additionally, the content of chrysolaminarin in diatoms is also affected by different culture conditions. It was shown that the crude chrysolaminarin in *P. tricornutum* was significantly increased under the P-deficiency and hypersaline conditions [37]. The chrysolaminarin in *T. pseudonana* and freshwater diatom *Asterionella formosa* was significantly decreased, while in *P. tricornutum* the chrysolaminarin content was significantly increased under high CO_2_ (20,000 ppm) [42]. The chrysolaminarin in fresh- and seawater *Navicula pelliculosa* did not have prominent difference under the high CO_2_. At the same time, nitrogen limitation resulted in the storage of β-1,3-glucan in *S. costatum* [43,44]. The content of β-1,3-glucan was up to 55% of the cell dry weight under the optimal cultivation conditions in *P. malhamensis* [10]. Silicon starvation and lower density culture led to the initial storage of chrysolaminarin in *T. pseudonana* [33]. This result could be explained by the fact that the immediate precursor, chrysolaminarin synthase and branching steps, cytoplasmic, and plastid gluconeogenesis gene transcripts were significantly upregulated at the first 4 h of silicon starvation. After 8 h of silicon starvation, the downregulation of glucanase genes resulted in the breakdown of chrysolaminarin and lipid storage as TAG [33]. Silicon limitation in *Cyclotella cryptica* did not change UGPase activity, decreased β-1,3-glucan synthase activity, and increased the activity of acetyl-CoA carboxylase [45]. The activity of these enzymes was proposed to be important for the storage of chrysolaminarin in diatom.

### 2.3. Bioactive Potentials of Laminarin

The bioactivities of chrysolaminarin were studied in some microalgae. For example, chrysolaminarin extracted from diatom *Odontella aurita* had a strong hydroxyl radical scavenging activity [17]. Besides, chrysolaminarin from freshwater diatom *Synedra acus* could suppress the growth and colony formation of human colon tumor cells [46]. The study of chrysolaminarin action mechanism on the cellular and molecular level showed that the biological action of chrysolaminarin was mediated by two membrane receptors, CR3 and Dectin-1 [46]. The chrysolaminarin of *P. malhamensis* could promote the fin regeneration of zebra fish by enhancing the antioxidant capacity of the injured zebrafish and reducing the potential damage of ROS to injured zebrafish [10]. It was verified that the bioactivities of chrysolaminarin are affected by their structure, molecular weight, solubility, and by their number of branch and helical conformation [10]. Therefore, different microalgal chrysolaminarins might have various bioactivities.

Furthermore, laminarin activity was more widely studied in brown algae. It was reported that laminarin extracted from *Laminaria digitata* could induce apoptosis in human colon and prostate cancers, and inhibited cell proliferation through activation of caspases via both death receptor-mediated and mitochondria-mediated apoptotic pathways [47]. Laminarin from brown algae acts as a facilitator of intestinal metabolism through changing the microbial community [48]. Detailed information about the effects of laminarin from brown seaweeds on anticancer were reviewed in a previous paper [49]. Additionally, it was also reported that laminarin plays important roles in anticoagulant, anti-inflammatory, immunoregulatory, antioxidant, and food applications as a functional ingredient [20].

## 3. Biosynthesis and Catabolism Pathways of Laminarin in Stramenopiles

### 3.1. The Proposed Pathways of Laminarin

The metabolic pathways of laminarin synthesis are poorly understood in stramenopiles. So far, the pathway was partially proposed in diatom *P. tricornutum* [50] and *T. psudonana* [33] and in alga *N. oceanica* [51], as shown in Figure 2. The core biosynthesis pathway of chrysolaminarin starts with the conversion of Glu-6-P to Glu-1-P via the catalysis of phosphoglucomutase (PGM). Five PtPGM genes were identified from the *P. tricornutum* genome, and two of them (PtPGM_1 (ID: 32708) and PtPGM_2 (ID: 48819)) showed diel expression patterns and contain chloroplast-targeting signals [41]. The Glu-1-P will be subsequently conversed to UDP-glucose under the UDP-glucose pyrophosphorylase (UGPase). Two putative PtUGPase genes (PtUGPase1 (ID: 50444) and PtUGPase2 (ID:23639)) were described in *P. tricornutum*. In *P. tricornutum*, PGM and UGPase are a fusion enzyme that is used to synthesize activated glucose. However, in addition to the PGM/UGPase fusion enzyme, an alternative pathway exists that uses an unfused UDP-glucose pyrophosphorylase in *N. gaditana* [13]. UDP-glucose is the major substrate for the chrysolaminarin biosynthesis [52]. Finally, two potential β-1,6-transglycosylases (PtTGS1 (ID: 50238) and PtTGS2 (ID: 56509)) involving in the branching of chrysolaminarin chain and a putative β-1,3-glucan synthase (PtBGS (ID: 56808)) for the backbone β-1,3-glucan synthesis were apparently located in the vacuoles of *P. tricornutum*, indicating the synthesis of chrysolaminarin might happen in the vacuole [50,52]. It was observed that PtUGP2, PtBGS and PtTGS1 had a coordinated diel expression pattern, with increase at the beginning and decrease at the end of the light period [41]. TGSs are very conserved among *N. gaditana*, yeast *S. cerevisiae,* and *P. tricornutum* [9].

In contrast, the chrysolaminarin is disassembled by endo-β-1,3-glucosidase and/or exo-β-1,3-glucosidase to UDP-glucose. In *P. tricornutum*, two endo-β-1,3-glucosidase (endo Glu1 (ID: 54681) and endo Glu2 (ID: 54973)) and three exo-β-1,3-glucosidase (exo Glu1 (ID:1372), exo Glu2 (ID: 45418) and exo Glu3 (ID: 49610)) with putative vacuolar localization were identified from its genome. Among them, two endo Glu genes and exo Glu1 showed a diel expression pattern with low expression during the light period and high expression during the dark period, while exo Glu2 showed conversely expression rule [41,53]. One putative β-glucan elicitor receptor (ID: 52685), a glucokinase (ID: 15495), and a Glu transporter (ID: 12520) may also be localized to the vacuole, showing higher expression during the dark period [41]. The glucokinase acts on the conversion of UDP-Glu to Glu-6-P during the glycolytic pathway, the Glu transporter may participate in the shuttle of Glu to the cytosol [41].

In *T. psudonana*, a PGM (Thaps3_35878) with putative plastid targeting and a fusion of PGM and UGPase enzyme (Thaps3_262059) with cytoplasmic localization were identified from the genome, indicating the generation of UDP-Glu might be in both compartments [33]. Three TGS genes (Thaps3_3105, 262361, and 263937) were found from the *T. psudonana* genome. Among them, Thaps3_3105 and Thaps3_262361 were potential targeted to periplastid (Thaps3_263937 in ER). One BGS (Thaps3_12685) did not have clearly predicted organellar targeting. Additionally, one exo Glu (Thaps3_13556) and two endo Glu (Thaps3_35711 and Thaps3_1554) with cytoplasmic localizations were identified from the genome. Although the subcellular localization of these proteins was predicted by bioinformatic methods, exact targeting still requires experiments to verify, such as eGFP fusion and fluorescence observed by confocal laser scanning microscope method, colocalization with organellar markers, and analysis via electron microscopy, as shown in our previous paper [54]. Anyway, it was proposed that the chrysolaminarin was stored in vacuole, owing to the vacuole-like aniline blue fluorescence in *T. psudonana*. 

The storage of polysaccharides mainly contains of four compartments in stramenopiles: (i) inside the plastid, (ii) in the cytoplasm, (iii) in vesicles surrounding the plastid cER (the outermost membrane of plastid) membrane, and (iv) in the vacuole [55]. In *P. tricornutum*, we found that chrysolaminarin located in vacuole via electron microscopy in our previous paper [18]. In *T. psudonana* and *N. oceanica* CCMP1779, it was proposed to target to vacuole and plastid, respectively [33,51]. Therefore, it needs experiments to certificate the exact localization of chrysolaminarin or laminarin in different stramnopiles.

During the light period, the fixed carbon through photosynthesis is totally transported to synthesize the nucleotide sugar UDP-Glu via the gluconeogenesis pathway in plastid. UDP-Glu is then shuttled to the vacuole, leading to the biosynthesis of chrysolaminarin. During the dark period, the chrysolaminarin is degraded to Glu and subsequently conversed to Glu-6-P for the glycolytic pathway. The genes involved in the chrysolaminarin biosynthesis were high expression during the light period and low expression during the dark period, while genes related with the catabolism of chrysolaminarin usually showed opposite expression pattern. Until now, the enzymes evolved in the conversion of Glu-6-P to UDP-Glu are predicted plastid localization via bioinformatic analysis, indicating that this process is performed in the plastid. While the enzymes related with the biosynthesis of chrysolaminarin from UDP-Glu are verified to vacuole localization, suggesting this process is carried out in vacuole. The catabolism of chrysolaminarin is also proposed to occur in vacuole by bioinformatic analysis.

### 3.2. Manipulation of Key Genes Involved in Chrysolaminarin Biosynthesis

Genes putatively involved in the biosynthesis of chrysolaminarin are identified from several *Nannochloropsis* genomes, but the functions of key genes (β-glucan synthase and transglycosylase genes) were only recently elucidated by experiments. It was shown that the knockout of these two genes resulted in an about five-fold decrease in soluble carbohydrate under the nitrogen deficiency, without an observed growth defect of the alga [13]. Similar results were also observed in *T. pseudonana*, where showed that the knockdown of β-1,3-glucan (chrysolaminarin) synthase gene (Thaps3_12695) decreased the accumulation of chrysolaminarin, and transiently increased TAG level with minimal detriment to growth [33]. 

The functions of key genes during the biosynthetic process of chrysolaminarin were also analyzed in *P. tricornutum*. The overexpression of phosphoglucomutase (PtPGM) in P. tricornutum significantly elevated chrysolaminarin content but reduced the lipid content, indicating the importance of PtPGM in regulating carbon flux [32]. UDP-glucose pyrophosphorylase (PtUGP) knockout mutant enhanced the production of lipid, especially the accumulation of triacylglycerol (TAG) [15]. It was also reported that PtUGP is an important rate-limiting enzyme during the chrysolaminarin biosynthesis, and the knockdown of this gene resulted in significantly lower chrysolaminarin content, but higher lipid content and slightly slower growth rate in *P. tricornutum* [56]. The expression of PtTGS genes complemented the growth deficiencies of yeast mutant, suggesting PtTGSs might be involved in the β-1,6-linked residues synthesis of chrysolaminarin [52]. Furthermore, the knockdown of PtBGS reallocated chrysolaminarin to soluble sugars and lipids inhibited their growth and photosynthetic capacities and increased their photoprotective abilities [50]. The complex intertwinement of different metabolites (polysaccharides, lipids and other carbon-containing metabolites) is important for the carbon balance in diatoms.

Although the functions of some genes involved in chrysolaminarin have already been studied, it is limited to a small number of microalgae and the biosynthetic pathway. Therefore, genes in more microalgae and involved in the catabolism of chrysolaminarin still require functional demonstration.

### 3.3. Diel Regulation of Chrysolaminarin

Diel regulation is an important environmental factor affecting photosynthetic autotrophic organisms. During the light period, photoautotrophs use solar energy to fix CO_2_ and drive anabolic processes for later bioprocesses during the dark period, such as respiration, cell division, stress sensitivity, chemotaxis, nutrient uptake [51,57]. Besides, the diel regulation plays a role in the physiological characterization of single photosynthetic autotrophic organisms, and also the communities and structure of phytoplankton [58].

Chrysolaminarin is a diurnal carbohydrate reserve compound [52]. Many studies showed that the accumulation of chrysolaminarin was a diurnal variation. The chrysolaminarin of diatom *S. costatum* and haptophyte *P. haptonemofera* was increased during the light period and decreased during the dark period, and it was explained by the elevated glucanase activity during the night [59]. At the same time, *N. oceanica* CCMP1779 was demonstrated to accumulate chrysolaminarin during the day and consuming it throughout the night [51]. Hildebrand et al., proposed that chrysolaminarin levels was a light dependence, the fixed carbon was initial storage as chrysolaminarin and then mobilized of that carbon into lipids and amino acids [33]. In *P. tricornutum*, the chrysolaminarin was accumulated during the day and near completely consumed in the dark [25]. It was already known that *P. tricornutum* cells are approximately divided once per day, with cell division beginning in the afternoon, continuing during the night, and finishing in the beginning of the next day [60]. Therefore, the accumulation of polysaccharides and lipids during the day provide energy for metabolisms and cell divisions at night [60]. The diel regulation of chrysolaminarin is not only observed in laboratory cultures of diatoms, but is also found in ocean. A recent study showed that the chrysolaminarin extracted from diatoms in surface water was diurnal turnover, affecting the elemental stoichiometry of the particle organic matter pool [8].

In stramenopiles, the global gene expression changed under diel cycles was only reported in *P. tricornutum* [41] and *N. oceanica* [51]. The genes involved in the biosynthesis of chrysolaminarin were high expressed during the light period, while genes involved in the metabolization were high during the night period. The diel expressed genes explained the diel regulation of chrysolaminarin biosynthesis in these microalgae.

## 4. Conclusions and Perspectives

To date, although the proposed biosynthesis and catabolism pathways of laminarin have already been explored, the knowledge regarding the functions of several genes have been extended. However, significant work remains to be done in order to characterize the complete biosynthesis and catabolism pathways and elucidate the functions of all key genes in microalgae (especially in the context of using microalgae as cell bio-factories to produce laminarin with different biological activities). Furthermore, the diel regulatory mechanism of laminarin still needs to be explored. Owing to the development of genome-editing tools in microalgae, genetic and metabolic engineering will be a potential method to study these microalgae and use them as cell bio-factories to produce large amounts laminarin with bioactivity. In brown algae, the studies mainly focused on the structure, content, and bioactivity of laminarin, while the pathways and genetic functions remain unknown due to the limited sequenced genome and genetic tools. 

Furthermore, some open questions are still required to be answered. For example: (1) how is the chrysolaminarin transported from the photosynthetic plastid to the storage vacuole; (2) how do the photosynthetic system and polysaccharide biosynthetic and mobilization pathways coordinate; and (3) what are the relationships among different carbon containing metabolites (e.g., polysaccharides, lipid, protein, and total carbon content)? Currently, biochemistry, cell biological, and molecular biological approaches will be used in our lab to try to answer these questions.

## Figures and Tables

**Figure 1 marinedrugs-19-00576-f001:**
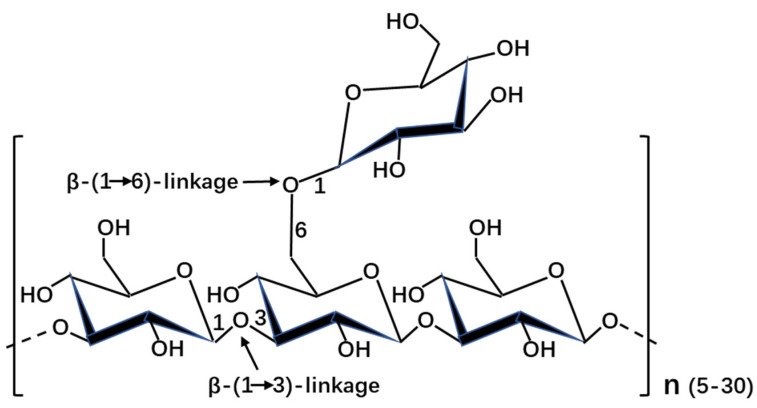
Schematic depiction of the structure of laminarin in stramenopiles.

**Figure 2 marinedrugs-19-00576-f002:**
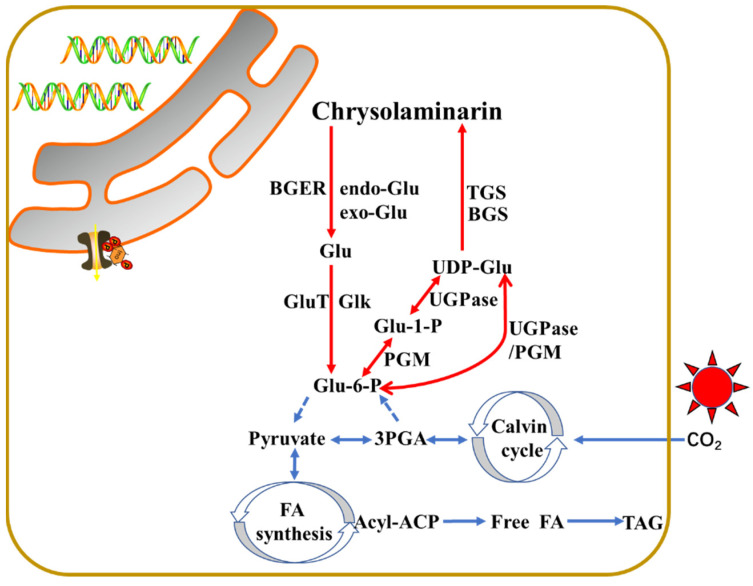
Schematic representation of the central synthesis pathway of chrysolaminarin in stramenopiles.

**Table 1 marinedrugs-19-00576-t001:** The structural features of laminarin in stramenopiles.

Species	Mw/DP	DB	Branches	Yield	Reference
*Phaeodactylum tricornutum*	nd		β-1,6	14%	[24]
*Phaeodactylum tricornutum*	DP 17	0.015	β-1,6		[25]
*Skeletonema costatum*	6–13kDa		β-1,6 β-1,2	32%	[26]
*Stauroneis amphixys*	4kDa DP~24		β-1,6 β-1,2	nd	[27]
*Achnanthes longipes*	nd		β-1,6 β-1,2	nd	[28]
*Craspedostauros australis*	>10 kDa		β-1,6	nd	[29]
*Aulacoseira baicalensis*	3–5 kDa		nd	0.9%	[11]
*Stephanodiscus meyerii*	40 kDa	0.053	β-1,6 β-1,3	0.5%	[30]
*Stephanodiscus meyerii*	2–6 kDa	0.25	β-1,6 β-1,3	0.4%	[30]
*Aulacoseira baicalensis*	nd	0.11	β-1,6 β-1,3 mannitol	0.6%	[11]
*Chaetoceros muelleri*	DP 22–24	0.006–0.009	β-1,6 β-1,3	nd	[16]
*Thalassiosira weissflogii*	DP 5–13		No branch	nd	[16]
*Chaetoceros debilis*	4.9 kDa, DP 30		β-1,6	10%	[31]
*Odontella aurita*	7.75 kDa,		β-1,6	15.09%	[17]
^a^ * Nannochloropsis gaditana*	DP 8	0.028–0.105	β-1,6 mannitol	0.5%	[9,13]
^b^ * Nannochloropsis gaditana*	DP 8.1–9.2	0.0036–0.0071	β-1,6 mannitol	0.5%	[9,13]
*Poterioochromonas malhamensis*	16.7 kDa		β-1,6	55%	[10]

Note: nd, not determined; DP, degree of polymerization; DB, degree of branching; Yield, the chrysolaminarin in % of diatom dry weight. ^a^: ^1^H-NMR analysis; ^b^: linkage analysis.

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
