# Peer review of "Laminarin, a Major Polysaccharide in Stramenopiles"

_marinedrugs, 2021, doi:10.3390/md19100576_

Round 1

Reviewer 1 Report

The subject of polysaccharides in stramenopile is exciting and valuable. 
However, the authors require to improve the MS on several points.
Introduction part: the polysaccharide termed "laminarin" written in "2. General features of ..." need to be explained first. For this reason, this part should be short.
As a familiar figure, showing laminarin as a figure is helpful for the reader's understanding.
All scientific names in the text need to be italic. 
In Table 1, the names of each genus written in P. S. A. C. T. O are difficult for readers to imagine. Please re-write the table.
Is Figure 1 your representative mentioned about the review? If it needs to be improved, please make the picture to be more impactful. The abbreviation is Glc, but some of the text's descriptions are Glu and UDP-Glu, and Glu-6-P.
Some of the IDs such as 54681, 49616 of endo-β-1,3-glucosidase are not distinguished from the web page.  Would you mind rechecking it?
The style of references is incomplete yet. Please put the end page of ref No 5, in addition, to be italic on the scientific name of 36.
Please correct the style of references on 1, 2, 3, 14, 16, 18, 28, 32, 33 and 52.
Overall, the MS is still incomplete yet. Would you please improve these minor points?

Author Response

The subject of polysaccharides in stramenopile is exciting and valuable.

However, the authors require to improve the MS on several points.

Introduction part: the polysaccharide termed "laminarin" written in "2. General features of ..." need to be explained first. For this reason, this part should be short.

Answer: We explained the laminarin and shorted this part as “β-1,3-glucans were mainly composed of glucose and also called laminarin, chrysolaminarin or mycolaminarin depending on algal species. β-1,3-glucan in brown algae was termed laminarin, in diatom Phaeodactylum tricornutum, Chrysophyte Poterioochromonas malhamensis and Eustigmatophyceae class Nannochloropsis gaditana was named chrysolaminarin [9-11,13], in oomycetes was named mycolaminarin [12]. These β-1,3-glucans are main carbohydrate molecule in the ocean carbon cycle and carbon pool [8].”

As a familiar figure, showing laminarin as a figure is helpful for the reader's understanding.

Answer: We added a depicted figure for laminarin, as shown in figure 1.

All scientific names in the text need to be italic.

Answer: We changed all scientific names into italic.

In Table 1, the names of each genus written in P. S. A. C. T. O are difficult for readers to imagine. Please re-write the table.

Answer: We re-wrote this table.

Is Figure 1 your representative mentioned about the review? If it needs to be improved, please make the picture to be more impactful. The abbreviation is Glc, but some of the text's descriptions are Glu and UDP-Glu, and Glu-6-P.

Answer: This figure is not the representative mentioned about the review, only for the central synthesis pathway of chrysolaminarin in stramenopiles. We checked all the Glc and Glu, made all consistent.

Some of the IDs such as 54681, 49616 of endo-β-1,3-glucosidase are not distinguished from the web page.  Would you mind rechecking it?

Answer: Yes, we rechecked them. All information corresponding to all IDs can be search on the Phaeodactylum tricornutum genome database. The link is as following: https://mycocosm.jgi.doe.gov/pages/search-for-genes.jsf?organism=Phatr2

The style of references is incomplete yet. Please put the end page of ref No 5, in addition, to be italic on the scientific name of 36.

Answer: We changed them as your suggestions.

Please correct the style of references on 1, 2, 3, 14, 16, 18, 28, 32, 33 and 52.

Answer: We corrected these references.

Overall, the MS is still incomplete yet. Would you please improve these minor points?

Answer: Yes, according to your scientific comments, we revised one by one carefully.

Reviewer 2 Report

Line 38-39: It would be beneficial to the reader if the authors provide a simple definition for stramenopiles with examples.

Section 2.1: Perhaps a general structure of laminarin should be discussed in the first paragraph followed by figures. Analytical methods can be elaborated in a later paragraph of the section. The section can start with what is currently the second or third paragraph.

Lines 104-108: Species names should be mentioned in italics. Please follow this rule throughout the document.

Line 133: Laminarin is usually measured using enzymatic methods which commonly employs kits from Megazyme. The authors have mentioned hot water extraction as a means of laminarin estimation. The reference study provided by the authors have however used a Megazyme kit. Hot water extraction (and alkali extraction) mentioned by the authors as estimation methods are laminarin extraction techniques. Since the authors are discussing measurement methods, they may as well include HPLC techniques devised by H. Zhang, K.H. Row, Extraction and separation of polysaccharides from laminaria japonica by size-exclusion chromatography.

Section 2.3: It would be beneficial for the reader if the authors explain the mechanism of bioactivities of laminarin such as apoptosis, fin regeneration etc.

Author Response

Line 38-39: It would be beneficial to the reader if the authors provide a simple definition for stramenopiles with examples.

Answer: We provided a simple definition for stramenopiles with examples. Among these Rhodophytes-derived lineages, stramenopiles with complex plastids are also called heterokonts, contain diatoms, giant macroalgae such as kelps, as well as photo-mixotrophic and heterotrophic species.

Section 2.1: Perhaps a general structure of laminarin should be discussed in the first paragraph followed by figures. Analytical methods can be elaborated in a later paragraph of the section. The section can start with what is currently the second or third paragraph.

Answer: Yes, we changed the paragraphs as your suggestions.

Lines 104-108: Species names should be mentioned in italics. Please follow this rule throughout the document.

Answer: We checked the line 104-108, and also the whole manuscript, change the species names to italics.

Line 133: Laminarin is usually measured using enzymatic methods which commonly employs kits from Megazyme. The authors have mentioned hot water extraction as a means of laminarin estimation. The reference study provided by the authors have however used a Megazyme kit. Hot water extraction (and alkali extraction) mentioned by the authors as estimation methods are laminarin extraction techniques. Since the authors are discussing measurement methods, they may as well include HPLC techniques devised by H. Zhang, K.H. Row, Extraction and separation of polysaccharides from laminaria japonica by size-exclusion chromatography.

Answer: We added HPLC technique into the manuscript.

Section 2.3: It would be beneficial for the reader if the authors explain the mechanism of bioactivities of laminarin such as apoptosis, fin regeneration etc.

Answer: We added the mechanisms of laminarin bioactivities into the manuscript (2.3), including on the colony formation of human colon tumor cells, the fin regeneration of zebrafish, the apoptosis in human colon and prostate cancers, and the facilitator of intestinal metabolism.

Round 2

Reviewer 1 Report

The authors responded to many reviewers' comments. 
The revised MS should be adequate to publish after minor corrections.
In the text, still "β-" and "beta-" are mixing. Please adjust the style to be "β-" as same as many other references. 

Author Response

Thanks very much for your suggestion. We changed all "beta" to "β".

Reviewer 2 Report

The article is good to be accepted in its present form.

Author Response

Thanks very much. We modified the manuscript as your suggestions.
